# STAT3 Activation in Psoriasis and Cancers

**DOI:** 10.3390/diagnostics11101903

**Published:** 2021-10-15

**Authors:** Megumi Kishimoto, Mayumi Komine, Miho Sashikawa-Kimura, Tuba Musarrat Ansary, Koji Kamiya, Junichi Sugai, Makiko Mieno, Hirotoshi Kawata, Ryutaro Sekimoto, Noriyoshi Fukushima, Mamitaro Ohtsuki

**Affiliations:** 1Department of Dermatology, Jichi Medical University, Shimotsuke-shi 329-0498, Tochigi, Japan; megumihkishimoto@gmail.com (M.K.); sashikawa@jichi.ac.jp (M.S.-K.); tuba2020@jichi.ac.jp (T.M.A.); m01023kk@jichi.ac.jp (K.K.); junsugar1112@gmail.com (J.S.); mamitaro@jichi.ac.jp (M.O.); 2Department of Medical Informatics, Center for Information, Jichi Medical University, Shimotsuke-shi 329-0498, Tochigi, Japan; mnaka@jichi.ac.jp; 3Department of Diagnostic Pathology, Jichi Medical University, Shimotsuke-shi 329-0498, Tochigi, Japan; kawata@jichi.ac.jp (H.K.); nfukushima@jichi.ac.jp (N.F.); 4Department of Pathology, Tokyo Metropolitan Cancer and Infectious Diseases Center Komagome Hospital, Bunkyo-ku, Tokyo 113-8677, Japan; rsekimoto@gmail.com

**Keywords:** psoriasis, STAT3, cancer, immunohistochemistry

## Abstract

Activation of signal transducer and activator of transcription (STAT)3 has been reported in many cancers. It is also well known that STAT3 is activated in skin lesions of psoriasis, a chronic skin disease. In this study, to ascertain whether patients with psoriasis have a predisposition to STAT3 activation, we examined phosphorylated STAT3 in cancer cells of psoriasis patients via immunohistochemistry. We selected patients with psoriasis who visited the Department of Dermatology, Jichi Medical University Hospital, from January 2000 to May 2015, and had a history of cancer. We performed immunostaining for phosphorylated STAT3 in tumor cells of five, four, and six cases of gastric, lung, and head and neck cancer, respectively. The results showed that there was no significant difference in STAT3 activation in any of the three cancer types between the psoriasis and control groups. Although this study presents limitations in its sample size and inconsistency in the histology and differentiation of the cancers, results suggest that psoriasis patients do not have a predisposition to STAT3 activation. Instead, STAT3 activation is intricately regulated by each disorder or cellular microenvironment in both cancer and psoriasis.

## 1. Introduction

Psoriasis is a chronic inflammatory skin disease associated with musculoskeletal symptoms in about 25% of patients [1]. Psoriasis is thought to be triggered by environmental factors such as trauma and infection in addition to genetic background, with both innate and acquired immunity involved in its pathogenesis. The pathogenesis of psoriasis has been intensively investigated, but its enigmatic nature has yet to be defined [2]. The relationship between psoriasis and STAT3 was first described by Sano et al. in 2005, when they reported that STAT3 was activated in keratinocytes of psoriasis lesions [3]. Since then, STAT3 hyperactivation has been reported in the cell types involved in psoriasis, including Th17 cells and keratinocytes [4].

STAT3 was first identified in 1993 [5] and known to be an important transcription factor and mediator in a number of different cell biological processes including proliferation, survival, differentiation, and angiogenesis under both physiological and pathological conditions [6,7]. It is one of the members of the seven STAT proteins, STAT 1, 2, 3, 4, 5A, 5B, and 6 [8]. Activation of STAT3 usually occurs through phosphorylation, in response to all IL-6 family members and various other cytokines, growth factors, oncoproteins, and hormones such as leptin [4,5,9]. STAT3 has two phosphorylation sites, namely a tyrosine residue (Tyr705) and a serine residue (Ser727) [10]. The canonical function of STAT3 as a transcription factor is mainly through the phosphorylation of tyrosine [7]. When a ligand binds to its cognate receptor, phosphorylation of Tyr705 occurs, resulting in dimerization of STAT3 and its translocation into the nucleus to exert its function as a transcription factor [5,11]. Non-receptor tyrosine kinases, such as c-Src, MAPK, and Abl are also involved in the activation of STAT3 through Tyr 705 phosphorylation [12]. Phosphorylation of Tyr705 plays a main and important role in the transcriptional function of STAT3, while phosphorylation of Ser727 also has various functions [12]. STAT3 is found in mitochondria, acting as a modulator of mitochondrial respiration and regulator of complex Ι activity and ROS production. These activities are related to the phosphorylation of Ser727 [13,14].

In recent years, there have been many reports of STAT3 overexpression has been found in cancer cells. Activation of STAT3 in malignant tumors has been implicated in poor prognosis, metastasis, and proliferation of cancers, and several STAT3 inhibitors are currently under development [5].

STAT3 is also involved in inflammation and immunity [15,16,17]. As mentioned above, Sano et al. reported in 2005 that STAT3 activation was observed in human epidermal keratinocytes in more than 90% (*n* = 19 of 21) of psoriatic lesions and some adjacent uninvolved epidermis by immunohistochemical analyses [3]. They also reported that transgenic mice expressing a constitutively active form of STAT3 in keratinocytes developed skin lesions that closely resembled human psoriasis [3]. STAT3 has recently emerged as a key player in the development and pathogenesis of psoriasis and psoriasis-like inflammatory conditions [4].

The risk of malignancy in patients with psoriasis is thought to be slightly increased compared to that in the normal population; thus, we speculated that STAT3 activation in psoriasis patients is related to malignancy risk.

In this study, we investigated the rate of active STAT3 tumors in patients with psoriasis compared to that in patients with eczema. To the best of our knowledge, very few studies have focused on STAT3 expression in the tumor cells of patients with psoriasis.

## 2. Materials and Methods

### 2.1. Patients

We selected Japanese patients with psoriasis and Japanese patients with eczema without psoriasis who presented to the Department of Dermatology in Jichi Medical University Hospital between 1 January 2000 and 31 May 2015. Among these patients, those with a medical history of non-skin cancers were selected for statistical analyses of malignancy risk. For the STAT3 immunohistochemical study, we extracted those who had undergone biopsy or surgery of their tumor at our hospital with a sufficient quantity of paraffin-embedded samples.

All patients with psoriasis and eczema were clinically diagnosed by experienced dermatologists with or without histological examination. All malignant tumors underwent histopathological diagnosis by pathologists. All patients were aged 20 years or above. All protocols were approved by the ethics committee of the Jichi Medical University.

### 2.2. Immunohistochemical Staining

Formalin-fixed, paraffin-embedded samples were sliced to a thickness of 5 mm. Antigen retrieval was performed by autoclaving sample slides at 120 °C for 10 min in citrate buffer (pH 6.0) and incubated with the primary antibody, rabbit monoclonal anti-phosphorylated STAT3 (Tyr 705) antibody (Cell Signaling Technology, Danvers, MA, USA), at a dilution of 1:100 overnight at 4 °C. Peroxidase staining was then performed with VECTASTAIN® ABC Kit (Vector Laboratories, Inc., Burlingame, CA, USA) following the manufacturer’s protocol, with diaminobenzidine (DAB, Dojindo, Kumamoto, Japan) as a chromogenic substrate.

### 2.3. Immunohistochemical Analysis

The outcomes of the staining were assessed by two factors: the staining intensity and the proportion of positive cells among cancer cells. Since phosphorylated STAT3 usually localizes to the nucleus, the staining of the nucleus was evaluated with anti-phosphorylated STAT3 antibody. The staining intensity was graded with a score of 0 to 3 (0: no staining, 1: mild staining, 2: moderate staining and 3: strong staining), and the proportion of positive cells was graded with a score of 0 to 3 (0: <1%, 1: 1–33%, 2: 34–66% and 3: 67–100%) by three independent experienced researchers under an optical microscope (BX53, Olympus, Tokyo, Japan).

The final score for each specimen was defined as the sum of the intensity score and proportion score.

### 2.4. Statistical Analysis

Chi-squared test and Student’s *t*-test were used to compare the groups. The Mantel-Haenszel method was used to determine the difference in the frequency of cancer in each organ between the psoriasis and control groups.

The Mann–Whitney U test was used to compare the expression of phosphorylated STAT3 between the groups. Statistical significance was set at *p* < 0.05. Statistical analysis was performed using IBM SPSS software (version 22.0).

## 3. Results

There was no significant difference in the frequency of malignant tumors in each organ between the psoriasis and eczema groups, but the frequency of patients with multiple malignant tumors was higher in the psoriasis group than in the eczema group.

A total of 103 cancers in 87 psoriasis patients and 135 cancers in 126 control patients were extracted from their medical records. The types of psoriasis patients included 79 cases (91%) of plaque psoriasis, one case (1.1%) of guttata psoriasis, four cases (4.6%) of generalized pustular psoriasis, one case (1.1%) of erythrodermic psoriasis, and two cases (2.3%) of psoriatic arthritis. This proportion is similar to that reported in the epidemiological surveillance of psoriasis patients in Japan from 2009 to 2012 [18]. Table 1 shows the cancer types in the psoriasis and control groups. There was no significant difference in the frequency of occurrence of cancers in any organ between the psoriasis and control groups, but the percentage of multiple cancers was significantly higher in the psoriasis patient group (*p* = 0.007). Among these, we selected cases with paraffin-embedded tumor samples that were able to match the cancer type with the control group.

These included five, four, and six cases in the psoriasis group and 16, six, and six cases in the control group, with gastric, lung, and head and neck cancers, respectively. The backgrounds of patients with psoriasis and control patients are shown in Table 2. All psoriasis patients selected for immunostaining had plaque type psoriasis, and three patients had double cancers: one with gastric cancer and lung cancer, one with lung cancer and lymphoma, and one with head and neck cancer and esophageal cancer. The histological type of each cancer was not completely matched among the groups, but sex ratio and age of onset of cancer were not statistically different.

The frequency of phosphorylated STAT3-positive cancers was not elevated in the psoriasis group compared to eczema group. Immunohistochemical staining images with anti-phosphorylated STAT3 antibodies representing the different scores are shown in Figure 1. There were no statistically significant differences in the staining scores of phosphorylated STAT3 between the psoriasis patient group and the control patient group for gastric, lung, and head and neck cancers.

## 4. Discussion

Many studies have shown that moderate to severe psoriasis are associated with comorbidities, such as cardiovascular disease, hypertension, metabolic syndrome, and psychiatric disorders. Cancer remains a matter of debate. Some cancers present no increased risk, but most studies have demonstrated the association of psoriasis with higher risks for cancer [19,20]. A recent meta-analysis in 2020 concluded that patients with psoriasis appear to have a slightly increased risk of cancer, particularly of keratinocyte cancer, lymphomas, lung cancer, and bladder cancer [21]. A systematic review in 2019 also showed that psoriasis was associated with an increased risk of overall cancer, as well as in site-specific cancers of the colon, colorectal, kidney, laryngeal, liver, lymphoma, keratinocyte, esophageal, oral cavity, and pancreatic cancer [20]. Our study demonstrated that there was no difference in the prevalence of cancer in psoriasis patients compared to eczema patients, but the frequency of patients with multiple cancers was significantly higher in the psoriasis group than in the eczema group, although the limitations of this study include the small sample size at a single institution and the fact that the histological type and grade of cancer were not consistent with those of the control group. Furthermore, the number of patients with sufficient samples at our hospital was limited because some patients underwent surgery at other institutions or did not undergo resection after diagnosis by biopsy. In order to compare psoriasis which is a Th17-balanced inflammatory skin disease, we chose the eczema group as a control, which is usually one of the Th2-balanced inflammatory skin diseases.

Psoriasis causes chronic low inflammation throughout the patient’s life, which increases the risk of malignancy. Psoriasis patients also have a higher ratio of smoking and/or alcohol consumption habits and higher body mass index (BMI), which would increase cancer risk. In addition, psoriasis treatment, such as systemic immunosuppressive drugs methotrexate and cyclosporine, and biologics may increase the risk of malignancy. In this study, we were not able to collect all the information on patients’ smoking and/or drinking habits, BMI, history of hepatitis B and/or hepatitis C, or their past use of immunosuppressive drugs that could influence the development of cancer. Thus, our data cannot determine whether the accurate risk of malignancies in psoriasis inflammation increased, but results showed that the overall frequency of psoriasis patients who developed cancers did not increase as the frequency of patients with multiple cancers increased, suggesting that the frequency of cancer-prone patients may be increased in the psoriasis population.

In previous reports, activation of STAT3 was detected in a wide variety of human cancer cells, including head and neck, brain, breast, gastric, colorectal, liver, lung, kidney, pancreas, prostate, ovarian, cervical cancer, multiple myeloma, and acute myeloid leukemia [22,23]. STAT3 is thought to be highly involved in cancer invasion, migration, metastasis, and angiogenesis, and plays an important role in cancer immune escape [22]. In relation, phosphorylation of Tyr705 and phosphorylation of Ser727 can then affect cancer metabolism [12]. With regard to gastric, lung, and head and neck cancers, which we examined in this study, previous reports showed that STAT3 activation was associated with negative factors such as poor prognosis of cancers [23,24,25].

In psoriasis, in addition to the activation of STAT3 in keratinocytes [3], STAT3 is activated by various stimuli in Th17 cells, which play an important role in the pathogenesis of psoriasis [4]. STAT3 is included as one of the genetic risk loci in psoriasis, and was reported as an up-regulated gene in a study of psoriatic patients compared to healthy controls [26,27]. Therefore, we designed this study to determine whether patients with psoriasis are susceptible to STAT3 activation in tumors. In this study, we investigated the activation of STAT3 in cancer cells by immunohistochemistry and found that the frequency of phosphorylated STAT3-positive cancers was not significantly different from that in the eczema group. 

In fact, phosphorylated STAT3 in cancer was mostly reported as a cancer promoter, but some studies indicated that it may also act as a suppressor under certain conditions [5]. In lung cancer, STAT3 was shown to play an unexpected tumor-suppressive role in *KRAS*-mutant lung adenocarcinoma [28]. High nuclear STAT3 expression levels are associated with favorable outcomes in head and neck squamous cell carcinomas [29]. Sano et al. reported that transgenic mice with keratinocytes expressing a constitutively active form of STAT3 developed psoriasis spontaneously [3], and that squamous cell carcinoma occurred early after carcinogenic stimuli in these mice [30]. Interestingly, in this transgenic mouse, squamous cell carcinoma avoided skin lesions of psoriasis [31]. These studies indicate that constitutive STAT3 activation in keratinocytes is involved in the pathogenesis of both psoriasis and skin squamous cell carcinoma, but oncogenic activation and inflammatory activation may differ.

Regarding its function in psoriasis, STAT3 activation in psoriatic keratinocytes occurs by IL-17, IL-19, IL-21, IL-22 [4], visfatin [32], and IL-36 [33]. These stimuli phosphorylate Tyr705 in STAT3. For example, activation of STAT3 by IL-22 is involved in the proliferation of keratinocytes [34], and activation of STAT3 by IL22 and IL-17A is involved in the induction of keratin 17, which is overexpressed in psoriasis [35,36,37]. Recently, it has been reported that oxidative stress caused by reactive oxygen species also promotes psoriasis through activation of STAT3 [38]. Ultraviolet B (UVB) activates STAT3 via phosphorylation of Tyr705 in the skin of mice. STAT3 activation was associated with a decreased UVB-induced apoptotic response and increased leukocyte infiltration and hyperplasia, suggesting a possible link to cancer [39]. In contrast, narrowband UVB irradiation had a suppressive effect on psoriasis by downregulating the expression of keratin 17 through inhibition of STAT3 activation, depending on the irradiation dose [40]. Another study in cultured keratinocytes showed that Jak2-dependent phosphorylation of Tyr705 induced by IL-6 and IL-20 resulted in a strong increase in the transcriptional activity of STAT3, and that ERK1/2- and p38 MAPK-dependent phosphorylation of Ser727 induced by tumor necrosis factor-α and UVB irradiation had a modulatory effect on the transcriptional activity of STAT3 [41]. Patients with psoriasis may have a genetic background that predisposes them to STAT3 activation [42]. The above studies suggest that differences in the mode of stimulation have different effects on STAT3 activation; that is, inflammatory stimuli from cytokines, such as IL-17 and IL-22, may activate STAT3 without influencing cancer risk, but oxidative stress, such as UV, may activate STAT3 with increased cancer risk.

## 5. Conclusions

The frequency of patients with psoriasis associated with cancer was similar to that of eczema patients, but the frequency of multiple cancers with psoriasis was increased compared to that with eczema patients. STAT3 is activated in psoriasis lesion and many cancers. STAT3 is a multifunctional protein whose function depends on the context of its activation. This means that there is a wide variety of stimuli and pathways that activate STAT3, and there are many different downstream reactions mediated by activated STAT3. STAT3 activation is observed both in psoriasis and cancers, however, STAT3 activation in keratinocytes involved in the pathogenesis of psoriasis; i.e., inflammatory STAT3 activation, may differ from oncogenic stimulation of STAT3 in cancers. The significance of STAT3 activation in inflammatory/oncogenic effects requires further investigation under specific conditions.

## Figures and Tables

**Figure 1 diagnostics-11-01903-f001:**
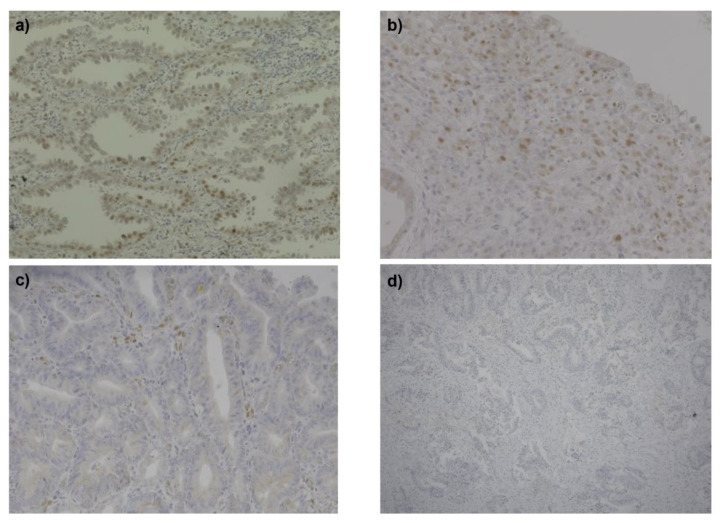
Staining images of phosphorylated STAT3 in cancer cells. (**a**) Expression of phosphorylated STAT3 in lung cancer developed by a control patient (score 5.33), scale bar=50 μm. (**b**) Expression of phosphorylated STAT3 in gastric cancer developed by a psoriasis patient (score 4), scale bar=50 μm. (**c**) Expression of phosphorylated STAT3 in gastric cancer developed by a psoriasis patient (score 3.33), scale bar=50 μm. (**d**) Expression of phosphorylated STAT3 in gastric cancer developed by a control patient (score 0), scale bar=100 μm.

**Table 1 diagnostics-11-01903-t001:** Types of cancer associated with psoriasis and control patients.

Types of Cancer	103 Cancers from 87 Psoriasis Patients			135 Cancers from 126 Control Patients		
		Male (*n* = 74)	Female (*n* = 13)		Male (*n* = 84)	Female (*n* = 42)
Gastric cancer	15	14	1	22	19	3
Liver cancer	15	13	2	14	12	2
Colon and rectal cancer	11	11	0	16	12	4
Lung cancer	10	10	0	13	9	4
Prostate cancer	8	8	0	13	13	0
Head and neck cancer	8	8	0	11	8	3
Renal cancer	6	4	2	3	2	1
Breast cancer	6	0	6	11	0	11
Esophageal cancer	5	5	0	2	2	0
Urinary tract cancer	5	5	0	7	5	2
Hematologic malignancy	5	5	0	8	4	4
Biliary tract cancer	3	2	1	1	1	0
Uterine cancer	2	0	2	7	0	7
Thyroid cancer	2	1	1	0	0	0
Pancreatic cancer	1	1	0	3	3	0
Mesenchymal tumor	1	1	0	2	2	0
Ovarian cancer	0	0	0	1	0	1
Cecal cancer	0	0	0	1	1	0
percentage of multiple cancers	19.5% *			7.7% *		

* *p* = 0.007.

**Table 2 diagnostics-11-01903-t002:** Patients’ characteristics.

	Psoriasis Patients				Control Patients			
	Number	Male/Female	Average Age of Onset	Histopathology	Number	Male/Female	Average Age of Onset	Histopathology
gastric cancer	5	4/1	69.8	adenocarcinoma 5	16	14/2	69.7	adenocarcinoma 16
lung cancer	4	3/1	63	squamous cell carcinoma 1 adenocarcinoma 3	6	5/1	66.3	small cell carcinoma 1 squamous cell carcinoma 1 adenocarcinoma 4
head and neck cancer	6	5/1	65.5	squamous cell carcinoma 6	6	4/2	69.0	squamous cell carcinoma 6

## Data Availability

Data sharing not applicable.

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
