# Peer review of "STAT3 Activation in Psoriasis and Cancers"

_diagnostics, 2021, doi:10.3390/diagnostics11101903_

Round 1

Reviewer 1 Report

Kishimoto et al. present an interesting study regarding STAT3 activation in patients with psoriasis and malignancies. The authors compared patients with psoriasis with malignancies to those with eczema with malignancies in a single center retrospective analysis. The results demonstrate that patients with psoriasis are more likely to have multiple malignancies than those with eczema, however there is no difference in STAT activation. This manuscript will be of interest to a specialized readership but requires some modifications:

Major Comments:

  1. What percentage of patients with eczema or psoriasis have malignancy in this center and how does this compare to other reports?
  2. Is eczema a good control group for this study? The authors should explain why they chose that comparator or choose additional disease controls.
  3. Which patients were included in the staining? Were they those with the highest likelihood of STAT3 activation based on tissue involvement or multiple malignancies? The authors should discuss.
  4. The authors should focus their discussion on studies with overlapping datasets instead of the broad areas of psoriasis and STAT3 activation.

Author Response

To reviewer 1

Thank you very much for your time to review our manuscript.

We have revised our manuscript following your comments.

Kishimoto et al. present an interesting study regarding STAT3 activation in patients with psoriasis and malignancies. The authors compared patients with psoriasis with malignancies to those with eczema with malignancies in a single center retrospective analysis. The results demonstrate that patients with psoriasis are more likely to have multiple malignancies than those with eczema, however there is no difference in STAT activation. This manuscript will be of interest to a specialized readership but requires some modifications:

Response: Thank you for your favorable comment.

Major Comments:

1. What percentage of patients with eczema or psoriasis have malignancy in this center and how does this compare to other reports?

Response: We did not evaluate the percentage of patients with malignancy in this study. We just picked up patients with psoriasis/or eczema who developed malignancy in their clinical course, and we do not know the percentage of patients with malignancy in this population.

2. Is eczema a good control group for this study? The authors should explain why they chose that comparator or choose additional disease controls.

Response: We chose eczema group as a control, because eczema is another category of inflammatory skin disease usually Th2 balanced, which is often compared to psoriasis, the Th17-balanced inflammatory skin disease. We added the following sentence on page 6, line 157. “In order to compare psoriasis which is a Th17-balanced inflammatory skin disease, we chose the eczema group as a control, which is usually one of the Th2-balanced inflammatory skin diseases.”

3. Which patients were included in the staining? Were they those with the highest likelihood of STAT3 activation based on tissue involvement or multiple malignancies? The authors should discuss.

Response: We stained all the samples which we were able to obtain from the pathology department. The sample was not arbitrarily selected; we chose patients for whom we had a sufficient amount of samples of malignant tumors in our hospital. In the first version of the article, we wrote "For STAT3 immunohistochemical study, we extracted those who had undergone biopsy or surgery of their tumor at our hospital with a sufficient quantity of paraffin-embedded samples.", but it was not clear, so we added the text below as limitations. On page 6, line 154 “The number of patients with sufficient samples at our hospital was limited because some patients underwent surgery at other institutions or did not undergo resection after diagnosis by biopsy.

4. The authors should focus their discussion on studies with overlapping datasets instead of the broad areas of psoriasis and STAT3 activation.

Response: We were not able to find the literature on studies with psoriasis, STAT3 and malignancy. Thus we discussed on psoriasis and STAT3, as far as we were able to obtain related literatures.

Reviewer 2 Report

The authors show that different types of cancer in psoriasis patients do not exhibit activation of STAT3 transcription factor compared with eczema control group. The paper is overall well written. Good quality antibodies were used for IHC. Several  remarks are listed below.

  1. The authors should better characterize psoriasis patients since the disease is classified into a number of types. Are there any relationships between activation of STAT3 and type of psoriasis, and/or the psoriasis area severity index (PASI)?
  2. Why the patients lacking any dermatological disorders were not used as a control? Are there any differences in the frequency of malignant tumors between eczema patients and healthy controls? The authors should clarify why eczema patients were used as a control group.

Author Response

To reviewer 2

Thank you very much for you time to review our manuscript. We revised our manuscript following your comments.

The authors show that different types of cancer in psoriasis patients do not exhibit activation of STAT3 transcription factor compared with eczema control group. The paper is overall well written. Good quality antibodies were used for IHC. Several remarks are listed below.

Response: Thank you for your favorable comments.

1. The authors should better characterize psoriasis patients since the disease is classified into a number of types. Are there any relationships between activation of STAT3 and type of psoriasis, and/or the psoriasis area severity index (PASI)?

Response: Thank you for your important comment. We completely agree that psoriasis should be more deeply characterized. We tried to classify the psoriasis patients into 5 categories, and compared the rates to that in overall psoriasis population. The following texts were added. On page 3, line 94, “The types of psoriasis patients included 79 cases (91%) of plaque psoriasis, 1 case (1.1%) of guttata psoriasis, 4 cases (4.6%) of generalized pustular psoriasis, 1 case (1.1%) of erythrodermic psoriasis, and 2 cases (2.3%) of psoriatic arthritis. This rate was not different from the rate reported in the epidemiological surveillance of psoriasis patients in Japan from 2009 to 2012” and on page 4, line 106, “All psoriasis patients selected for immunostaining had plaque type psoriasis, and three patients had double cancers: one with gastric cancer and lung cancer, one with lung cancer and lymphoma, and one with head and neck cancer and esophageal cancer.” As for the relationship between STAT3 activation and the type of psoriasis, we could not compare the relationship between the disease type and STAT3 activation because all of the cases for which immunostaining was performed had plaque psoriasis. These patients developed malignancies during their long history of psoriasis, which naturally wax and wane, and also with treatments, and it is very difficult to evaluate their disease severity, such as PASI.  

2. Why the patients lacking any dermatological disorders were not used as a control? Are there any differences in the frequency of malignant tumors between eczema patients and healthy controls? The authors should clarify why eczema patients were used as a control group.

Response: It was technically easier for us to chose eczema patients as a control group than to choose non-dermatological patients as a control group. More than that, eczema is usually contrasted to psoriasis because of its Th2-balanced nature. We added the following sentence on page 6, line 157. “In order to compare psoriasis which is a Th17-balanced inflammatory skin disease, we chose the eczema group as a control, which is usually one of the Th2-balanced inflammatory skin diseases.”

Round 2

Reviewer 1 Report

The authors have responded appropriately to the reviewers comments.